# The Central Nervous System Source Modulates Microglia Function and Morphology In Vitro

**DOI:** 10.3390/ijms24097685

**Published:** 2023-04-22

**Authors:** Andreia G. Pinho, Andreia Monteiro, Sara Fernandes, Nídia de Sousa, António J. Salgado, Nuno A. Silva, Susana Monteiro

**Affiliations:** 1Life and Health Sciences Research Institute (ICVS), School of Medicine, University of Minho, Campus de Gualtar, 4710-057 Braga, Portugal; id9529@alunos.uminho.pt (A.G.P.); b13265@med.uminho.pt (A.M.); nidiadesousa@med.uminho.pt (N.d.S.); asalgado@med.uminho.pt (A.J.S.); nunosilva@med.uminho.pt (N.A.S.); 2ICVS/3B’s—PT Government Associate Laboratory, 4710-057 Braga, Portugal

**Keywords:** microglia, cortex, spinal cord, morphology, phagocytosis, in vitro studies

## Abstract

The regional heterogeneity of microglia was first described a century ago by Pio del Rio Hortega. Currently, new information on microglia heterogeneity throughout central nervous system (CNS) regions is being revealed by high-throughput techniques. It remains unclear whether these spatial specificities translate into different microglial behaviors in vitro. We cultured microglia isolated from the cortex and spinal cord and analyzed the effect of the CNS spatial source on behavior in vitro by applying the same experimental protocol and culture conditions. We analyzed the microglial cell numbers, function, and morphology and found a distinctive in vitro phenotype. We found that microglia were present in higher numbers in the spinal-cord-derived glial cultures, presenting different expressions of inflammatory genes and a lower phagocytosis rate under basal conditions or after activation with LPS and IFN-γ. Morphologically, the cortical microglial cells were more complex and presented longer ramifications, which were also observed in vivo in CX3CR1^+/GFP^ transgenic reporter mice. Collectively, our data demonstrated that microglial behavior in vitro is defined according to specific spatial characteristics acquired by the tissue. Thus, our study highlights the importance of microglia as a source of CNS for in vitro studies.

## 1. Introduction

Rio-Hortega first described the microglia in 1919 as a distinct population within the central nervous system (CNS) [1,2]. These cells adopt several morphological states throughout their lifetime and display high migratory and phagocytic activities under pathological conditions [1,2,3,4]. For almost a century, researchers have considered microglial cells as bystanders of CNS physiology with the sole purpose of clearing CNS debris. However, in recent years, insights into the complex biology of these cells have raised relevant questions about their functions in health and disease [5]. Currently, microglia are known as resident immune cells of the CNS that play important roles in the development and homeostasis of the brain and spinal cord and respond to immune challenges in pathological or injury contexts [6,7,8,9].

Studies of mouse models concerning the origin of microglia revealed that during development, microglial progenitors arise from uncommitted CD31^+^ C-KIT^+^ erythromyeloid precursor (EMP) cells within the embryonic yolk sac (YS). These cells migrate into the brain rudiment at embryonic day 9.5 (E9.5), where they are established and maintain their self-renewal capacity during each year of fetal development [10,11,12]. Once established, embryonic microglia expand and colonize the entire CNS until adulthood [11]. Recent studies suggested that the spatial heterogeneity of microglia may be influenced by the surrounding cellular microenvironment and different developmental stages, contributing to the morphological and functional differences in microglia within the CNS in health and disease. It is important to note that although it is still unclear which intrinsic and extrinsic factors shape microglia properties, it is plausible that microglia, when migrating to other areas of the CNS, such as the cortex and spinal cord, may be exposed to different maturation processes, leading to distinct CNS microglia signatures [13,14]. Indeed, in vitro studies have demonstrated that cortex and spinal cord microglial cells present distinct protein signatures and biological properties and that this differential heterogeneity is present throughout the lifespan and after pathogen exposure [14,15,16]. For instance, microglia from the spinal cord have been shown to be involved in inflammatory processes, whereas microglia from the cortex play a crucial role in neuronal migration [14].

Several studies have indicated that microglial behavior can be highly dynamic and shaped according to spatial and context (disease) determinants [15,17,18,19,20]. However, microglia isolated from the cortex are still commonly used as a gold-standard in vitro model to study neuroinflammation. In this study, we focused on further investigating the differences in the activation profile, phagocytosis capacity, and morphology of cortical versus spinal cord microglia in vitro. The data provided herein support the current evidence showing that even in the same microenvironment, microglia isolated from the cortex differ from spinal cord microglia in vitro. Importantly, we showed that these morphological differences observed in vitro are also present in vivo.

This study highlights the need to carefully select the origin of isolated microglia according to the research question so as to avoid drawing biased conclusions.

## 2. Results

### 2.1. Cortex vs. Spinal Cord Microglia Present Distinct Activation Profiles In Vitro

Microglia-distinct behaviors have been documented across central nervous system regions and pathological contexts [13,15,17,20]. It is unclear whether this in vivo heterogeneous behavior translates into distinct functionality on the in vitro level. Conclusions from experiments using cortex glial or enriched microglial cultures have been applied transversely to brain- and spinal-cord-based studies, which may constitute a misguided analysis. We conducted an unbiased in vitro study using glial cultures from cortex and spinal cord tissue isolated from the same animal with the same experimental protocol (Figure 1A). After establishing the glial culture, we analyzed the inflammatory profile by examining non-activated (basal conditions) and activated (cells incubated for 24 h with LPS and INF-γ) cultures (Figure 1A). As a first readout, we examined the microglial cell numbers using immunocytochemistry (Figure 1B). Although no differences were found in the basal state, the number of microglia was higher in the activated spinal cord cultures (Figure 1C, spinal cord: 214.2 ± 24.75) when compared to the activated cortical cultures (Figure 1C, cortex: 116.5 ± 19.39), revealing a distinctive response of these cells when in a pro-inflammatory state (Figure 1C). Multiple comparison post hoc analysis confirmed that the number of microglia cells was significantly different according to CNS origin (F(1,14) = 7.429 *p* = 0.0164) and activation (F(1,14) = 4.923, *p* = 0.0435). To consolidate these data, we assessed the same parameters using flow cytometry. The analysis demonstrated that the concentration of CD11b+ cells in the spinal cord cultures was significantly higher (Figure 1D spinal cord: 312,691 ± 44,691) than that in the cortical glial cultures under activated conditions (Figure 1D, cortex: 156,427 ± 6272, Figure 1D,E, F(1,14) = 19.90, *p* = 0.0003). To further dissect the activation profile of the microglial cells, we verified the expression of iNOS, CD86, and CD200R in these cells (Figure 1F–K). Flow cytometry analysis of iNOS expression revealed that the glial cultures derived from spinal cord tissue presented a higher percentage of microglia expressing iNOS (Figure 1G spinal cord: 8.642 ± 0.3470) when in an activated state compared to the cortex-derived glial cultures (Figure 1G, cortex: 7.092 ± 0.4405, Figure 1F,G, F(1,20) = 7.271, *p* = 0.0139). Moreover, it is important to highlight the contrast in the expression of this marker between the non-activated and activated glial cultures (Figure 1F,G, F(1,20) = 695.8, *p* < 0.0001). The interaction of both factors, CNS origin and activation, revealed that CNS sources modulate the behavior of these cells in vitro, causing them to react differently to the same stimuli (Figure 1F,G, F(1,20) = 7.156, *p* = 0.0145). Interestingly, both CD86 and CD200R were differentially expressed between the spinal-cord- and cortex-derived cultures under both basal and activated conditions and were highly expressed in cells derived from the cortex (Figure 1I, spinal cord in activated conditions: 14.00 ± 2.238 and cortex: 26.88 ± 1.518, Figure 1F,H–K; F(1,20) = 77.90, *p* < 0.0001 and F(1,20) = 63.82, *p* < 0.0001, respectively).

### 2.2. Microglia Phagocytosis Function In Vitro Differs according to The Primary CNS Source

As previously mentioned, microglia significantly contribute to central nervous system function by clearing apoptotic and cell debris through phagocytosis during development, homeostasis, and disease paradigms [21]. We followed the same experimental design to analyze how microglial function is altered in vitro according to the region of origin (Figure 2A). After establishing the glial culture, we incubated the cells with GFP fluorescent microspheres and evaluated the phagocytosis rate and cytokine production in non-activated and activated microglia (Figure 2A). Surprisingly, this process was revealed to be dynamic between non-activated and activated conditions, revealing a significant effect of factor interaction (CNS origin and activation) (Figure 2B, F(1,11) = 12.72, *p* = 0.0044). Under basal conditions, microglia derived from the spinal cord presented less phagocytosis than microglia isolated from the cortex (Figure 2B spinal cord: 39.05%, ±1.089 compared to cortex: 50.46%, ±4.767 *p* = 0.0428). Conversely, after activation, the spinal cord microglia showed an increased phagocytosis function (to 46.58%, ±1.369) compared to the cortex microglia phagocytosis rate, which was significantly decreased (to 37.33%, ±1.963 *p* = 0.0406) (Figure 2B). We also evaluated microglial phagocytosis by flow cytometry, which allowed us to analyze the presence of engulfed beads, specifically in the microglia, through the previous gating of this population (Appendix A). This quantification again showed an effect of the CNS origin on microglial phagocytosis function (Figure 2C, F(1,19) = 62.22, *p* < 0.0001), where, in accordance with the immunocytochemistry data, spinal-cord-derived microglia in the basal state (Figure 2C, spinal cord: 18.25 ± 1.637) had a lower phagocytosis rate (compared to cortex: 32.75 ± 1.670) (Figure 2B,C). However, in the activated glial cultures, spinal cord microglia phagocytosis (Figure 2C, spinal cord: 19.70 ± 2.143) was significantly lower than cortex microglia phagocytosis (Figure 2C, cortex: 32.02 ± 1.389), *p* = 0.0004). The supernatants of these cultures were used to characterize the cytokine panel released before and after incubation with the fluorescent beads (Figure 2D–F). Curiously, the production of IL-23 was influenced by incubation with the fluorescent beads, with the IL-23 levels were below the limit of detection (LOD) before incubation and highly increased after bead incubation (Figure 2D, F(1,15) = 160.2, *p* < 0.0001). Moreover, the cortex-derived glia produced more IL-23 when incubated with fluorescent beads under basal conditions (Figure 2D cortex:0.2789 ± 0.004995) than the spinal-cord-derived glia (Figure 2D, spinal cord: 0.1850 ± 0.04937, Figure 2D, F(1,15) = 11.97, *p* = 0.0035), which, in addition to being a tendency, was maintained after the activation protocol (Figure 2D, *p* = 0.0629). IL-18 was produced in slightly higher amounts after activation; however, no statistically significant difference was observed between the groups (Figure 2E). The CXCL1 cytokine followed the same profile, being produced in high amounts after the activation protocol for both the cortex- and spinal-cord-derived cultures, independent of fluorescent bead incubation (Figure 2F, F(1,15) = 212.8, *p* < 0.0001). Moreover, in the activated cultures, a CNS origin effect was noted, with the cortex-derived cells being greater producers of CXCL1 than the spinal-cord-derived cells (Figure 2F, cortex: 1.390 ± 0.05250 vs. spinal cord: 0.6206 ± 0.1359, Figure 2F, F(1,15) = 31.80, *p* < 0.0001).

### 2.3. Microglia Derived from Cortex vs. Spinal Cord Present Distinct Morphology In Vitro and In Vivo

Microglia, which are immune cells of the central nervous system, are engaged in constant surveillance of the surrounding microenvironment, presenting striking morphological plasticity. Additionally, it is known that microglia morphology and function are closely related [22]. Considering the functional differences in microglia, we sought to compare the cortex and spinal microglial morphologies in glial cultures. For this purpose, we followed the same rationale as the experiments presented above and evaluated only the non-activated microglial cells (Figure 3A), since it has already been established how microglial morphology changes after a pro-inflammatory stimulus. Therefore, after imaging the IBA-1 staining of both glial cultures, Sholl analysis of the individual microglial cells was performed (Figure 3B). Interestingly, cortical microglial cells showed increased intersecting ramifications, revealing a more complex cell morphology than that of spinal cord microglial cells, with fewer ramifications and a shorter cell radius (Figure 3C, F(1,396) = 32.73, *p* < 0.0001). Taking advantage of CX3CR1^+/GFP^ transgenic mice, we evaluated these microglial morphological differences in vivo by acquiring high-resolution representative images of microglia in the cortex and spinal cord tissues (Figure 3D). Importantly, we compared only cells located in the gray matter so that our data were not influenced by the previously described differences between microglia on white and gray matter [19]. The Sholl analysis of individual CX3CR1-GFP^+^ cells (Figure 3E) revealed that the cortical microglia presented a more complex morphology than the spinal cord cells, with a higher number of intersections (Figure 3F, F(1,234) = 117.3, *p* < 0.0001).

## 3. Discussion

Microglial heterogeneity has been extensively studied in vivo using high-throughput techniques. However, how exactly these differences observed in vivo are maintained in vitro and how they affect microglial behavior is still yet to be fully understood. Moreover, it is crucial to use suitable in vitro models to understand and manipulate microglial functions to study microglia in healthy and diseased conditions. However, cortex-derived microglia are still commonly used as the gold-standard in vitro model to study distinct CNS injuries or disorders [23,24,25,26,27].

This study compared microglial activation, phagocytosis, and morphology in cultures obtained from cortex or spinal cord tissues. To perform an accurate comparison, we isolated the cells from these two regions (cortex and spinal cord) from the same P7 Wistar rat and followed the same protocol (Figure 1A, Figure 2A and Figure 3A). This procedure controls for confounding factors, such as species, sex, genetic background, and age. 

Cortex- and spinal-cord-derived glial cultures, under basal conditions, presented similar numbers of microglia cells (Figure 1C,E). However, a slight tendency towards higher IBA-1^+^ cell or CD11b concentrations was observed for in the spinal cord cultures (Figure 1B–E). This can be explained by greater microglial representation in the original CNS tissue. However, in addition to being reported as a difference in regional density throughout the CNS, the number of microglia in the spinal cord is lower than that in the cortex [13,28]. Another hypothesis for the tendentially greater number of microglia in spinal-cord-derived cultures is a higher survival rate or, for example, more proliferation in vitro. Proliferation is a common reaction of microglia to immune stimuli. One possibility is that the in vitro environment promotes spinal cord microglia activation. In fact, the in vivo environment differs from in vitro conditions and can shape the activation profile. Additionally, the tissue isolation protocol can alter the microglia phenotype. However, when searching for morphological or phenotypic correlates of activation possibly caused by the in vitro environment or technical procedures, we observed that the microglia from cultures that were not exposed to our activation protocol exhibited a ramified morphology (1B) and did not express the iNOS marker (Figure 1G), suggesting a non-activated phenotype. Thus, we then evaluated whether microglia from the spinal cord were more reactive towards an inflammatory challenge. We incubated microglial cells with LPS and IFN-γ, known stimuli inducing cell proliferation [29] and nitric oxide biosynthesis [30]. Curiously, after 24 h of stimulation, the glial cultures derived from spinal cord cells showed a significantly higher number of microglial cells than the glial cultures from the cortex (Figure 1C,E). This observation, confirmed using immunocytochemistry and flow cytometry, showed that in response to the same stimuli, microglia derived from cortex and spinal cord tissues react differently, reaching higher numbers in spinal cord cultures. Again, these higher microglial densities can be explained by a higher proliferation rate related to a different capacity to react to the activation stimulus. It would be interesting to analyze whether there is a different expression of LPS receptors (TLR4) in the microglia from each CNS source or if this may even be a feature acquired in vitro. When evaluating iNOS expression in these cells, a marker commonly upregulated upon activation, the stimulatory effect was confirmed in both cultures, with a higher percentage of CD11b^+^iNOS^+^ cells after activation (Figure 1G). Second, a CNS effect was observed, with a higher percentage of iNOS^+^ microglial cells present in the spinal-cord-derived culture (Figure 1F,G). Interestingly, when observing CD86, a membrane co-stimulatory receptor responsible for T cell activation and proliferation [31], and CD200R, an immune receptor involved in cytokine production [32], only a CNS origin effect was observed (Figure 1H–K). Thus, spinal-cord-derived cells presented lower levels of CD86 and CD200R in non-activated and activated conditions (Figure 1H–K), showing that although this specific activation protocol did not induce this immune profile, it is possible to confirm a CNS origin effect on microglia signatures in vitro. Collectively, these data demonstrate a distinct microglial activation profile in vitro according to the primary CNS source, which is in line with previous data from the Haas lab, reporting that CD86, as is the case for other immunoregulatory proteins, presents a region-specific CNS expression [33]. More recently, the Aymerich S. M. group described different expressions of CD86 across distinct brain regions, including the cortex, hippocampus, and striatum, with a higher expression of this activation marker in comparison to the midbrain [34]. Although differential microglial CD86 expression has already been described in distinct brain regions and on the in vivo level [17,33,35], our data show that these differences may now have to be considered in in vitro studies.

In addition to their capacity to respond to and activate inflammatory cascades, microglia also regulate CNS homeostasis through phagocytosis [7]. To study whether phagocytosis function is also affected in vitro according to distinct CNS origins, we followed the same rationale as that used before with additional incubation using fluorescent YG carboxylate microspheres (Figure 2A). Immunocytochemistry showed that, in a basal state, microglia isolated from the spinal cord had less phagocytic capacity than cells isolated from the cortex (Figure 2B). In the activated cultures, the microglial phagocytic capacity was altered, with the spinal-cord-derived cells presenting a slightly higher phagocytosis rate than the cortex-derived cells (Figure 2B). Although these phagocytic dynamics are interesting, we are aware that immunofluorescence analysis of ingested beads may carry some errors, since some of the counted beads may only adhere to the cell surface. 

As an alternative, we performed flow cytometry, which enables the discrimination of cells according to complexity and size parameters, reducing the counting bias (Appendix A). In this analysis, we did not observe a significant effect of the activation protocol (Figure 2C). However, the flow cytometry data support the observation that the phagocytosis capacity of microglia in vitro is distinct according to their CNS spatial origin, as the spinal-cord-derived microglia are less phagocytic (Figure 2C). It is important to highlight that environmental cues modulate microglial transformation from a surveying to phagocytic function differently [36], and the effect observed with microspheres should also be tested with other engulfed substrates present in the CNS tissue context, such as synaptosomes, myelin debris, or apoptotic cells [37,38,39,40]. Here, we observed that the impact of unique signals from the cortex or spinal cord microenvironment on microglial cell machinery is most likely conserved in vitro, resulting in different phagocytic capacities. 

We then analyzed the microglial cytokine profile according to the CNS origin in response to phagocytosis or the activation protocol. The production of IL-23 was significantly increased after incubation with fluorescent beads under both basal and activated conditions. This is in line with previous reports stating that phagocytes secrete this cytokine, which has mostly been described for neutrophils and peripheral macrophages [41,42,43]. However, although it has been described that microglial cells are producers of IL-23 [44] and that this cytokine plays roles in different CNS pathologies [45,46], to the best of our knowledge, we still lack a connection between this cytokine and microglial phagocytosis. Moreover, since microglia also express the IL-23 receptor [45], it is possible that this cytokine acts in an autocrine manner by regulating microglial functions such as phagocytosis. Interestingly, our data show a CNS-origin-dependent effect on IL-23 levels that is consistent after inducing microglial phagocytosis, being observed in the spinal-cord-derived glial cultures; lower IL-23 production (Figure 2D); and a lower phagocytosis rate (Figure 2B,C). Based on these data, it would be relevant to further explore the possible association between IL-23 and microglial phagocytosis. The possible association between IL-23 and microglial phagocytosis raises new hypotheses concerning the region-specific accumulation of plaques in regions of demyelination and neuronal loss observed in CNS autoimmune pathologies [47,48,49,50]. 

The signaling molecule IL-18 was detected to only a slight degree after the activation protocol (Figure 2E), which is consistent with its association with microglial pro-inflammatory processes [51,52,53,54,55]. This activation effect was also detected on CXCL1 release, which was produced in high amounts after the activation protocol, independent of the presence of phagocytosis beads (Figure 2F). The CXCL1 profile is associated with its function as a ligand involved in the microglial pro-inflammatory profile following several CNS inflammation processes [56,57]. In addition to the activation effect, for CXCL1, it is also possible to confirm a CNS-region-specific response, with spinal-cord-derived cells being lesser producers of this chemokine in comparison to cells present in cortex-derived cultures (Figure 2F). To our knowledge, no studies have addressed the CNS-region-specific expression and release of CXCL1.

Overall, our cytokine/chemokine analysis demonstrated that the CNS source of glial cells affects the production of IL-23 and CXCL1 in vitro, with spinal cord glial cells exhibiting lower production. Importantly, this analysis may lack specificity for the microglial response, since it was performed on mixed glial culture supernatants, meaning that the molecules analyzed could have been released by microglia but also by oligodendrocytes or astrocytes [49,58,59].

In addition to their impressive variety of functions described previously, microglia can easily adapt to a new context/microenvironment through morphological changes. In fact, the analysis of cell morphology in the field of microglia research has been widely employed, in some cases being associated with the immune profile—the so-called morphology phenotype. There are several classifications of ramified, amoeboid, hypertrophic, rod, dystrophic, satellite, gitter-cell-like, or dark microglia, all trying to correlate morphology with functionality [60]. Moreover, microglia vary in morphology depending on their location, a fact known since 1990, and have been extensively studied across different CNS regions to date [28]. Thus, microglial morphological characterization has been performed on different brain regions [61,62] and spinal cord tissues, mainly after injury [63]. 

Considering our previous functional differences, we questioned whether cell morphology was also altered in vitro depending on the original tissue location. To address this, we analyzed the morphology of individual microglial cells (Figure 3A,B). We observed a difference in the number of intersections along the cell radius between the cortex and spinal cord microglial cells in cultures, where the cortex-derived microglia were longer and more complex (Figure 3C). Regional differences in microglial morphology between the brain and spinal cord have been reported previously [13]. Curiously, this contrast is particularly striking in gray and white matter tracts, where microglia range from 5% to 12% of the total cells per region, respectively, with higher densities found in the gray matter [64]. Since the cortex and spinal cord tissues present different proportions of white and gray matter, this could explain the observed microglial morphological differences. To overcome this concern and clarify whether the microglial morphology already differs in the original location, we took CX3CR1^+/GFP^ mice and isolated cortex and spinal cord tissues from the same animal (Figure 3D). High-resolution images of the gray matter regions alone were taken for the Sholl analysis (Figure 3E). An equivalent result was observed in vivo, showing that microglia from the cortex had a complex morphology and presented longer ramifications (Figure 3F). Therefore, our in vivo analysis supports the differences observed in vitro. The shorter and less ramified morphology of spinal cord microglia coupled with the higher numbers previously verified in the culture and increased expression of iNOS suggest that these cells, in culture, are more reactive to activation.

In contrast to this hypothesis, Jesudasan et al. revealed that spinal cord microglia exhibit a less inflammatory phenotype and a less amoeboid morphology when compared with brain microglia in response to LPS [15]. However, this data divergence can easily be explained by the different animal ages (postnatal day 1 vs. day 7 in our study), protocol specificities regarding the culture medium (DMEM-F12 vs. DMEM), and activation components (1μg LPS vs. 10 ng/mL ng/mL LPS + 20 ng/mL IFN-γ). More recently, Murgoci et al. showed that cortex and spinal cord microglia cultured in vitro have distinct protein signatures and biological properties [14]. Specifically, they observed that microglia from the spinal cord in the context of injury are involved in inflammatory processes, while microglia from the cortex play a crucial role in neuronal migration and exogenesis [14]. Importantly, both studies showed that microglial behavior differs in vitro according to the original spatial location in the central nervous system. Moreover, in a study using the amyotrophic lateral sclerosis (ALS) model, Nikodemova et al. reported regional heterogeneity in the microglial phenotype and function, even within the affected regions, where the microglia appear to be more reactive and macrophage-like than cortex microglia [17]. Lastly, a recent study using scRNA-seq to characterize microglial heterogeneity in WT mice and transgenic HIV-1 gp120 mice revealed overlapping but distinct microglial populations in the cortex and spinal cord. They found two clusters of microglia, homeostatic microglia (HOM-M) and inflammatory microglia (IFLAM-M), in distinct proportions, revealing that the cortex may have a more limited capacity for a microglia-mediated inflammatory response. In vitro models have been developed to predict in vivo conditions reliably and efficiently. Our data show that microglia differ throughout the CNS and demonstrate several differences between microglia on the in vitro level depending on their original tissue source. 

## 4. Materials and Methods

### 4.1. Study Design

P7 newborn Wistar rats of both sexes (Charles River) (RRID_RGD_737929) were used for in vitro experiments. Adult (10–15 weeks; 23–28 g) male CX3CR1^+/GFP^ (RRID:IMSR_JAX:005582) (Charles River) mice were used for the in vivo analysis of microglia morphology. In total, 45 animals were used. For the in vitro studies, 7 animals per experiment were used in a total of 6 independent experiments. For in vivo tissue analysis, 3 animals were used. The animals used in this study were maintained under standard laboratory conditions (12 h light/12 h dark cycle, 22 °C, relative humidity of 55%, ad libitum access to standard food and water). Rats were maintained in numbers of 2 per cage and mice 5–6 per cage in enriched cages with paper. All experiments were approved by the Portuguese Regulatory Entity (DGAV 022405) and conducted in accordance with the local regulations on animal care and experimentation (European Union Directive 2010/63/EU).

### 4.2. Cortex and Spinal Cord Glial Cultures

Cortical and spinal cord glial cells were isolated from P7 newborn Wistar rats (RRID_RGD_737929) as described previously [65]. Briefly, the rats were euthanized by decapitation, and the meninges were removed by dissection of the cortical and spinal cord tissues. The tissues were kept separated. The cortices and spinal cords were enzymatically digested in a dissociation solution (30 mg/mL DNase 30 (Sigma, Saint Louis, MO, USA, Cat. No. 9003-98-9), 0.25% trypsin (Gibco, Billings, MT, USA, Cat. No. 25300062)) for 30 min at 37 °C, followed by mechanical dissociation. This dissociation procedure allows for a better separation of cells from the surrounding tissue and simultaneously increases neuronal cell death, which is known to be sensitive to these procedures. Enzymatic digestion was stopped by adding 40% newborn calf serum (NBCS) (Cat. No. 26010074, Invitrogen, Waltham, MA, USA) to the cell suspension, which was further centrifuged at 800 rpm for 2 min to obtain glial cells. The glial cell pellet was mechanically resuspended in culture medium (DMEM; Sigma, Saint Louis, MO, USA, Cat. No. D5648) supplemented with NaHCO_3_ (Sigma, Saint Louis, MO, USA, Cat. No. S5761), 10% newborn calf serum, and 1% penicillin-streptomycin (P/S, Gibco, Billings, MT, USA, Cat. No. 15070063) using 5 and 10 mL pipettes. A second centrifugation was performed at 1 200 rpm for 5 min to remove the tissue debris. The pellet was resuspended in culture medium, and a 1:1 dilution of Trypan Blue (Sigma, Saint Louis, MO, USA, Cat. No. T8154) was prepared for cell counting under a light microscope, the Olympus BX51WI Fixed Stage Upright Microscope (RRID:SCR_023069), using a Neubauer chamber (Marienfeld, Harsewinkel, Germany, Cat. No. 0640010). A density of 50,000 cells/well was used for a 24-well plate or 1,000,000 cells/well for a 6-well plate for immunocytochemistry or cytometry analysis, respectively. The cells were maintained on culture medium at 37 °C with 5% CO_2_ for 21 days in vitro (DIV), and the medium was changed every 3 days.

### 4.3. Microglia Activation Response and Phagocytosis Function Assays

Confluent glial cultures from the cortex or spinal cord were obtained after 21 DIV, and their activation response and phagocytosis capacity were assessed. IFN-γ (20 ng/mL, PeproTech, Cat. No. 400-20) and LPS (10 ng/mL, Sigma, Saint Louis, MO, USA, Cat. No. L4391) were added to the culture medium to study the activation response. Half of the wells were stimulated in each condition (activated glial culture), while the other half were maintained with basal culture medium (non-activated glial culture). The glial cells were stimulated for 24 h at 37 °C with 5% CO_2_. The activated and non-activated cells were incubated with 0.0025% (*v*/*v*) fluorescent YG carboxylate microspheres (Polysciences, Inc., USA, Cat. No. 15700-10) with a diameter 0.5 μm for 1 h at 37 °C with 5% CO_2_ to study the phagocytic function.

### 4.4. Microglia Immunocytochemistry

The cortical and spinal cord microglia were analyzed using immunocytochemistry after activation and phagocytosis assays. The cells were fixed in 4% paraformaldehyde for 30 min at room temperature (RT), and membrane permeabilization was performed with 0.2% Triton-X in PBS (PBS-T) for 5 min at RT. The cells were then blocked with 10% NBCS in PBS-T for 30 min at RT, followed by a 60 min incubation with primary antibodies, namely, rabbit anti-IBA-1 (diluted 1:1000 in 10% NBCS/PBS, Wako, Osaka, Japan, Cat. No. 019-19741) to detect microglial cells and mouse anti-iNOS (diluted 1:100 in 10% NBCS/PBS, Abcam Cat# ab15323, RRID:AB_301857) to analyze the microglial activation phenotype. Primary antibody incubation was omitted to produce negative controls. The secondary antibodies, Alexa Fluor 594 anti-rabbit (diluted 1:1000 in 10% NBCS/PBS, Thermo Fisher Scientific Cat# A-11037, RRID:AB_2534095) and Alexa Fluor 488 anti-mouse (diluted 1:1000 in 10% NBCS/PBS, Thermo Fisher Scientific Cat# A-11034, RRID:AB_2576217), were incubated in the dark for 60 min at RT. DAPI (diluted 1:1000 in 10% NBCS in PBS; 1μg/mL, Invitrogen, Waltham, MA, USA, Cat. No. 62248) treated for 5 min at RT was used for nuclear staining. The wells were maintained in PBS and protected from light until the time of fluorescence microscopy analysis. Microglia in the IBA-1^+^ cells were analyzed, in addition to microglia activated by IBA-1^+^iNOS^+^ and phagocytosis activated by IBA-1^+^GFP^+^ cells. All histological procedures and evaluation were performed blind to the experimental groups.

### 4.5. Flow Cytometry Analysis

After the activation or phagocytosis assays, the glial cells were detached from the wells using 0,05% trypsin (Gibco, Billings, MT, USA, Cat. No. 25300062) for 3 min at 37 °C. Trypsinization was stopped by doubling the volume of culture medium. The cells were centrifuged at 1200 rpm for 5 min, and the pellet was resuspended in DMEM. A 1:1 dilution of Trypan Blue (Sigma, Saint Louis, MO, USA, Cat. No. T8154) was prepared for cell counting under a light microscope, the Olympus BX51WI Fixed Stage Upright Microscope (RRID:SCR_023069), using a Neubauer Chamber (Marienfeld, Harsewinkel, Germany, Cat. No. 0640010). Flow cytometry staining was performed as previously described [66] with a total of 1 × 10^6^ cells. For membrane staining, glial cells were resuspended in flow cytometry staining buffer (FACSb: phosphate-buffered saline (PBS), 10% BSA (Sigma, Saint Louis, MO, USA, Cat. No. A-9418), 0.1% sodium azide) and then incubated in the dark for 20 min at 4 °C with the following primary antibodies: PECy7 anti-rat CD11b/c (BioLegend Cat# 201817, RRID:AB_2565946), APC anti-rat CD86 (BioLegend Cat# 200315, RRID:AB_2910355), and FITC anti-rat CD200R (BioLegend Cat# 204905, RRID:AB_2074190). 

For intracellular staining, glial cells were fixed in 2% PFA for 20 min at 4 °C followed by one round of 200 rpm centrifugation for 2 min at 4 °C. Permeabilization was performed for 5 min at 4 °C using a permeabilization buffer solution of 1× (diluted from 10× stock with water). After centrifugation at 1200 rpm for 2 min at 4 °C, primary staining was performed using the primary antibody, rabbit anti-iNOS (diluted 1:100 with permeabilization buffer solution, Abcam Cat# ab15323, RRID:AB_301857), for 20 min at 4 °C, followed by incubation for 10 min at 4 °C with the secondary antibody, Alexa Fluor 488 anti-rabbit (diluted 1:1000 in permeabilization buffer solution 1×) (Thermo Fisher Scientific Cat# A-11034, RRID:AB_2576217). The cells were acquired using a BD LSRII Flow Cytometer (BD LSR II Flow Cytometer (RRID:SCR_002159)). 

Doublets were excluded by FSC-A versus FSC-H, and dead cells were excluded by positive staining for 7-AAD (BioLegend, Cat. No. 420403) (Appendix A). Microglial cells were gated according to CD11b^+^ expression (Appendix A). Activated microglia were analyzed according to the positive expression of CD11b^+^CD86^+^, CD11b^+^CD200R^+^ (Appendix A), and CD11b+iNOS+ (Appendix A). Phagocytic microglia were analyzed based on the positive expression of CD11b and GFP (Appendix A). Sample acquisition and gating analysis were performed blind to the experimental groups.

### 4.6. LEGENDplex^TM^ Multi-Analyte Flow Assay of Glia Culture Supernatants

The cultured glial cell supernatants were assayed using the LEGENDplexTM Mouse Macrophage/Microglia Panel kit (Biolegend, Cat. No. 740846) according to the manufacturer’s instructions. This kit includes the following analytes: CXCL1 (KC), Free Active TGF-β1, IL-18, IL-23, CCL22 (MDC), IL-10, IL-12p70, IL-6, TNF-α, G-CSF, CCL17 (TARC), IL-12p40, and IL-1β. Briefly, reagents were prepared from the stocks provided, and standard serial dilutions were prepared to generate a standard curve. Assay buffer (25 µL) was added to the standard and sample wells at a 1:1 ratio. Then, 25 μL of mixed beads was added to each well, and the plate was incubated for 2 h at RT with continuous agitation at 800 rpm. After centrifugation at 250 g for 5 min, the beads were washed with 1× wash buffer for 1 min. Detection antibodies (25 µL) were added to each well, followed by 1 h of incubation at RT with agitation at 800 rpm. Streptavidin-phycoerythrin (SA-PE) (25 µL) was added directly to the previous solution, and the plate was incubated for 30 min at RT with agitation at 800 rpm. After washing with 150 μL of 1× wash buffer, the samples were ready to read using the flow cytometer. The samples were vortexed, and 300 beads per analyte were acquired using a cytometer (BD LSR II Flow Cytometer (RRID:SCR_002159)).

The FCS files were analyzed using Biolegend’s LEGENDplex^TM^ data analysis software suite. The concentration of each detected analyte was presented in pg/mL. Analytes with concentrations below the detection limit (LOD) in all the samples were excluded from the analysis. Analytes that were partly detected in some samples were analyzed and plotted as graphs that included the LOD representation as a dashed line. The experimenter was blind to the experimental groups when performing the protocol, sample acquisition, and analysis. 

### 4.7. Microglia Morphology Analysis

Since microglial morphology has been related to the cell functional state, this parameter was also analyzed by comparing the cortical and spinal cord microglia on both the in vitro and in vivo levels. The morphology was analyzed using the Sholl analysis plugin of the Fiji software from Fiji (RRID:SCR_002285) [67]. 

For in vitro analysis, IBA-1+ cells from the cortex and spinal cord glial cultures (see the protocol in the first section of the Materials and Methods) were imaged under an Olympus BX61 Upright Wide-Field Microscope (RRID:SCR_020343). 

For the in vivo analysis, we used transgenic adult (10–15 weeks; 23–28 g) male CX3CR1^+/GFP^ (RRID:IMSR_JAX:005582) mice. In this reporter mouse, CX3CR1+ cells, such as monocytes, subsets of natural killer cells, and microglia cells, express enhanced green fluorescent protein (eGFP). Adult heterozygous CX3CR1-GFP mice (3 months of age) were anesthetized with an intraperitoneal injection of a mixture containing ketamine (Imalgene, 75 mg/kg, Merial, Lyon, France) and medetomidine (Dormitor, 1 mg/kg, Pfizer, New York, NY, USA) to avoid animal suffering and intracardially perfused with saline followed by 4% PFA. The cortex and spinal cord tissues were dissected. The tissues were sectioned in a cryostat, and transversal 20 μm thick sections were produced and collected on slides. Slides containing the cortex and spinal cord sections were placed and rehydrated in PBS (1×) at room temperature. Finally, the tissues were mounted onto Immu-Mount^®^ (Thermo Scientific, Waltham, MA, USA, Cat. No. FIS9990402), and representative images were taken using an Olympus FV1000 Confocal Microscope (RRID:SCR_020337) (Olympus, Hamburg, Germany).

Both in vitro and in vivo microphotograph analyses were performed, as previously described [68]. Briefly, the images were converted into 8 bits and thresholded, and the largest Sholl radius was defined by creating a straight line starting at the center of the analysis. Concentric rings spaced 5 μm apart were used to calculate the number of intersections in each radius, as shown in Figure 3. The selection of microglia for analysis was performed blind to the experimental groups.

### 4.8. Statistical Analysis

Statistical analysis was performed using GraphPadPrism ver.8.0 (RRID:SCR_002798). One-way ANOVA followed by Tukey’s test was used to compare four groups. When comparing two factors, we used two-way ANOVA followed by Sidak’s multiple comparisons test. Normality was measured using the Kolmogorov–Smirnov and Shapiro–Wilk statistical tests. Equality of variances was measured using Levene’s test and was assumed when *p* > 0.05. Values were accepted as significant if *p* < 0.05. Data are presented as the group mean ± standard error of the mean (SEM).

## 5. Conclusions

In summary, we reported that the origin of microglia is relevant in determining the behavior of these cells in vitro. Specifically, our results revealed that microglia in spinal-cord-derived cultures are functionally and morphologically different from those in cortex-derived cultures. Thus, the region selected for the isolation of microglia cells will influence the cultures accordingly and affect defined outcomes. Therefore, it is essential to align the research question with the adequate in vitro experimental design.

## Figures and Tables

**Figure 1 ijms-24-07685-f001:**
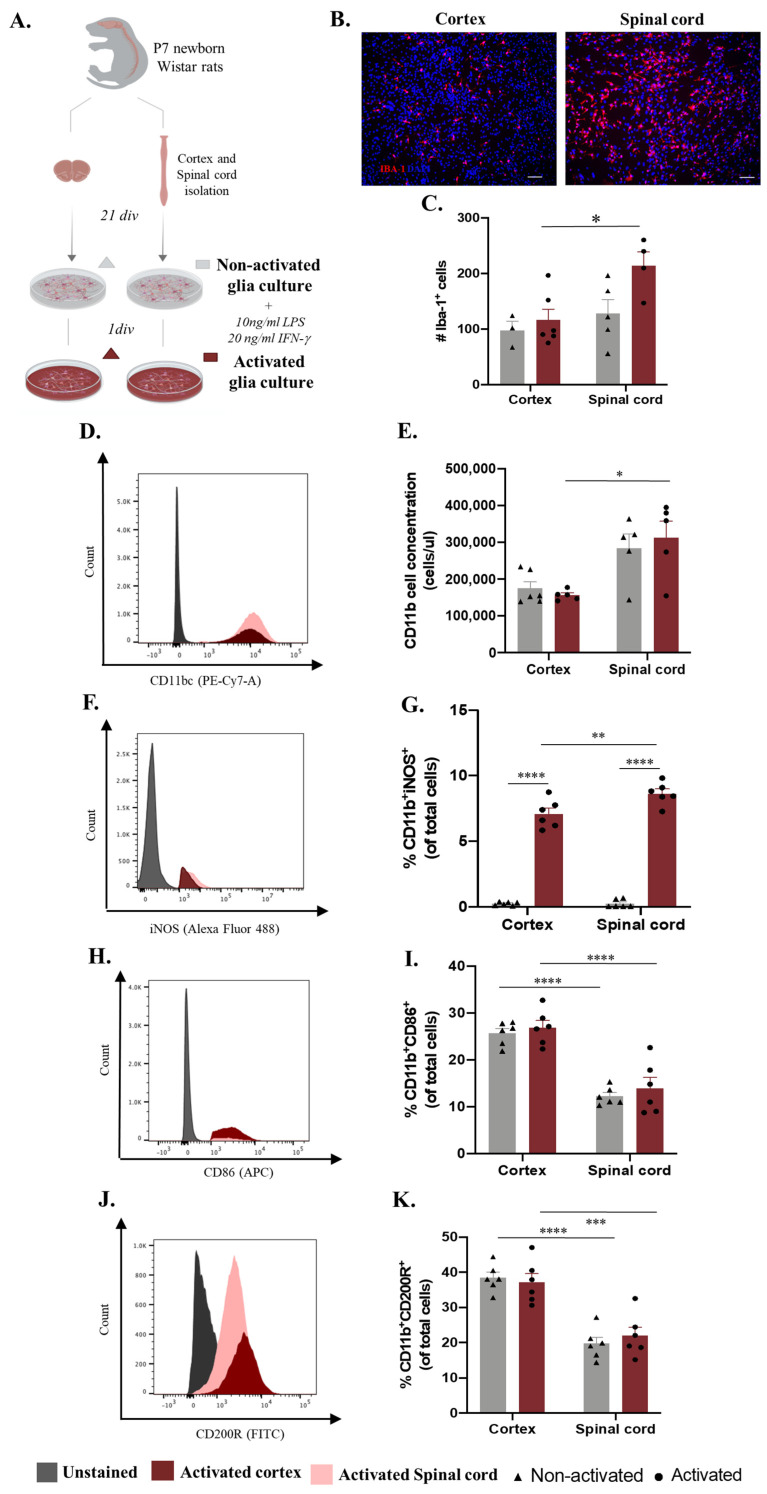
Microglia differences in vitro according to central nervous system (CNS) spatial origin. (**A**) Experimental setup of the in vitro protocol used to compare microglia from the cortex and spinal cord in both non-activated and activated conditions. (**B**) Representative images of non-activated cortex and spinal cord glial cultures. Scale bar: 100 μm. (**C**) Quantification of the IBA-1^+^ cells in basal and activated conditions (**D**). Representative histogram of CD11b/c population in cortex cultures (highlighted in red) and spinal cord (highlighted in pink). (**E**) Cell concentration of microglia in glial cultures in basal and activated conditions. (**F**) Representative histogram of iNOS expression in activated cortex (highlighted in red) and spinal cord (highlighted in pink) microglial cells. (**G**) Percentage of microglia iNOS+ in basal and activated conditions. (**H**) Representative histogram of CD86 expression in activated cortex cultures (highlighted in red) and spinal cord (highlighted in pink) microglial cells. (**I**) Percentage of microglia CD86+ in basal and activated conditions. (**J**) Representative histogram of CD200R expression in activated cortex (highlighted in red) and spinal cord (highlighted in pink) microglial cells. (**K**) Percentage of microglia CD200R+ in basal and activated conditions. *n* = 3–6, number of independent cell culture wells. Results expressed as mean ± SEM. * *p* <0.05; ** *p* <0.01, *** *p* < 0.001, and **** *p* < 0.0001.

**Figure 2 ijms-24-07685-f002:**
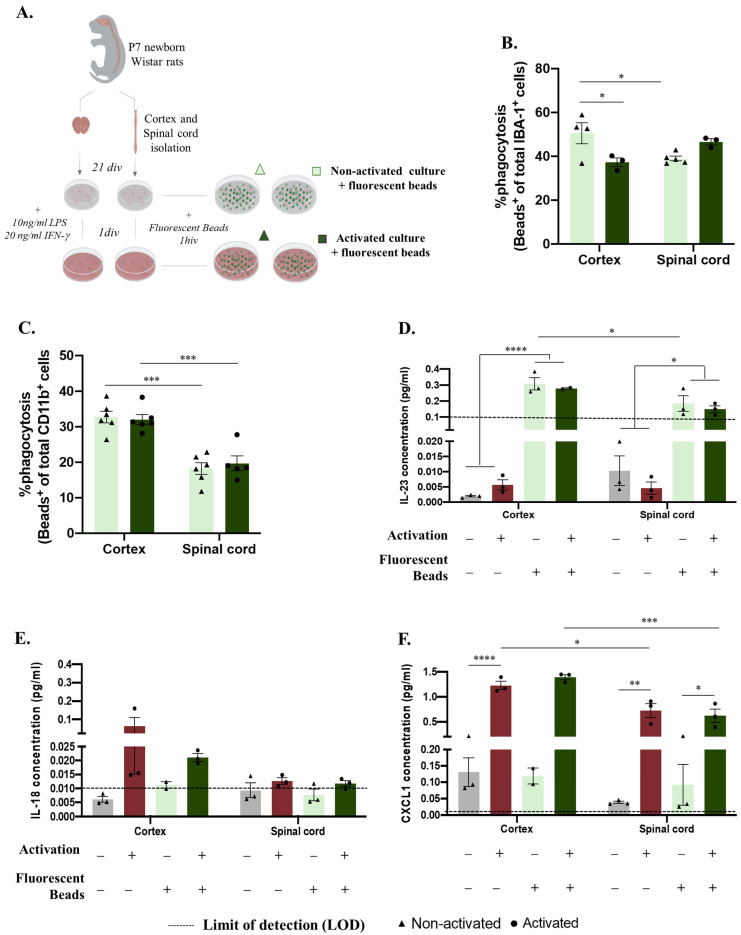
Central nervous system spatial origin alters microglia phagocytosis in vitro. (**A**). Experimental setup of the in vitro protocol used to compare microglia phagocytosis (**B**). Microglia phagocytosis rate quantification by immunocytochemistry in basal and activated culture conditions (**C**). Microglia phagocytosis rate determination by flow cytometry in glial cultures in activated and non-activated conditions. (**D**) IL-23 concentration in basal and activated glial culture supernatants before and after incubation with GFP microspheres. (**E**) IL-18 concentration in basal and activated glial culture supernatants before and after incubation with GFP microspheres. (**F**) CXCL1 concentrations of the different paradigms. Limit of detection in cytokine analysis: 0.1065 (IL-23), 0.0138 (IL-18), and 0.0044 (CXCL1). *n* = 3–6, number of independent cell culture wells. Results expressed as mean ± SEM. * *p* < 0.05, ** *p* < 0.01, *** *p* < 0.001, and **** *p* < 0.0001.

**Figure 3 ijms-24-07685-f003:**
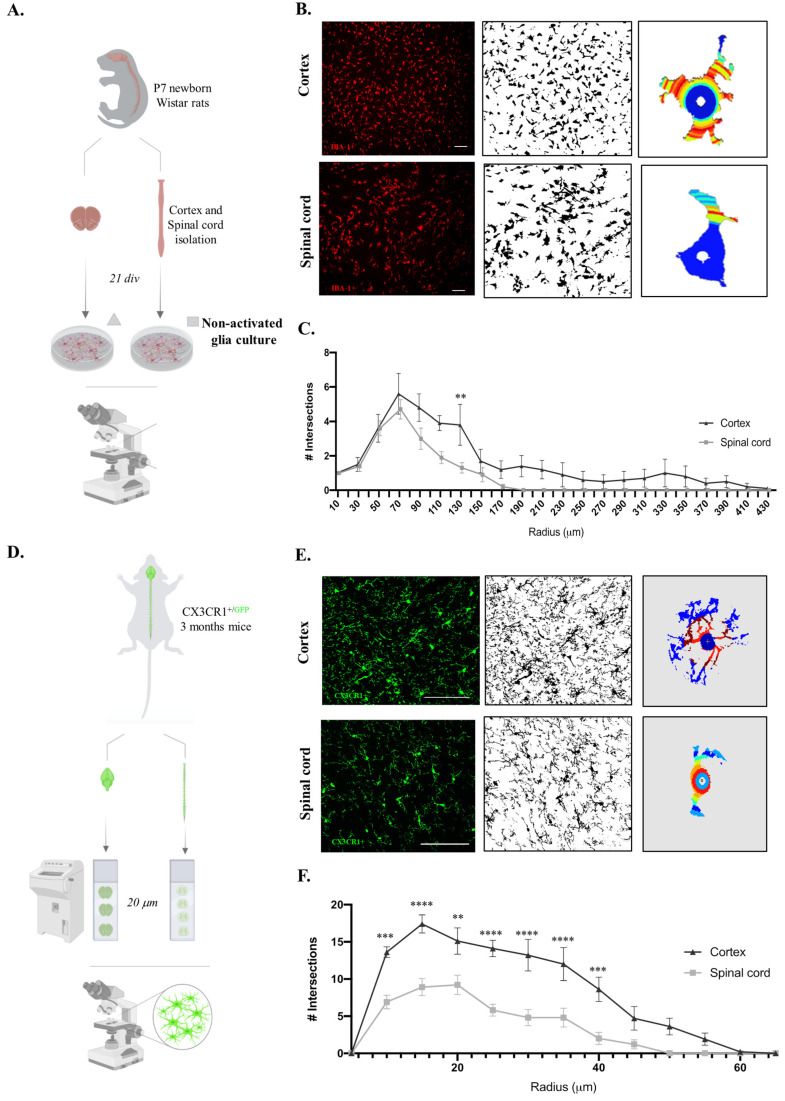
Cortex microglial cells are morphologically more complex than spinal cord microglia on the in vitro and in vivo levels. (**A**) Experimental setup of the in vitro protocol used to compare microglia morphologies. (**B**) Sholl analysis protocol steps for the acquired microscope images of IBA-1 staining, binary conversion with Fiji software (version:2.0.0-rc-65/1.49v), and application of Sholl analysis plugin to individual microglial cells. Scale bar in white represents 100 μm. (**C**) Quantification of microglial cell intersections. (**D**) Experimental setup of the in vivo protocol used to compare microglia morphologies. (**E**) Sholl analysis protocol steps for the acquired confocal images of CX3CR1-GFP^+^ cells, binary conversion with Fiji software, and application of Sholl analysis plugin to individual microglial cells. Scale bar in white represents 100 μm. (**F**) Quantification of microglial cell intersections. *n* = 10, number of individual cells analyzed. Results expressed as mean ± SEM. ** *p* < 0.01, *** *p* < 0.001, and **** *p* < 0.0001.

## Data Availability

The datasets used and/or analyzed during this study are available on reasonable request from the corresponding author.

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
