# Peer review of "The Central Nervous System Source Modulates Microglia Function and Morphology In Vitro"

_ijms, 2023, doi:10.3390/ijms24097685_

Round 1
Reviewer 1 Report
The authors of this manuscript investigated the regional heterogeneity of microglia at an in vitro level, isolating cells from the cortex and spinal cord. They found that microglial behavior could be highly dynamic and shaped according to specific spatial characteristics acquired in the tissue. This is an instructive area to navigate and worth understanding in more detail. However, there are still some issues in the article that need to be addressed.
1. Writing and formatting errors: There are errors and mistakes in writing and formatting of this article, which reduce the rigor and readability of the study. For example, Figure 1 lacks "A" and "B" icons; "Fig 1F" at line 94 and "Fig. 1F" at line 96 are inconsistent; the meaning of "SC" at line 130 is unclear. Please revise the article format more carefully to ensure consistency and clarity.
2. Figure 1C concerns: In Figure 1C, the authors claim that the number of microglia in the spinal cord culture group increased significantly after stimulation compared to the cortical culture group. However, there are differences in experimental replication between the two groups, and there is an obvious deviation from the mean value in the spinal cord culture group at the basal state. The authors should address whether this situation could lead to statistical differences after stimulation and provide a clearer explanation.
3. Non-activated state: The in vivo microenvironment is different from the in vitro situation, and primary microglia isolated from the cortex and spinal cord can be easily activated. The authors should provide evidence to support their claim that the "non-activated state" is genuine. This is especially relevant for the results in Fig. 1I and 1K, where no statistically significant difference can be observed between the basal state and stimulation group. A more in-depth explanation or additional data would be helpful to clarify this point.
4. Figure 2 inconsistencies: In Figure 2, the statistical results are expressed as mean 卤 SEM, but the data presented at lines 130-133 only show the mean values. Additionally, the statistical graph patterns in Fig. 2B-C differ from those in Fig. 2D-F. In Fig. 2E, the authors claim that "no statistically significant difference was observed between the groups." To make the results more reliable, the authors should provide the raw data and ensure consistency in the presentation of statistical results throughout the figure.
5. Image resolution issues: The resolution of the immunofluorescence images is too low to accurately identify cell staining. The authors should provide higher-resolution images to allow for proper evaluation of the staining patterns and results.

Reviewer 2 Report
The study of Pinho et al. analyzes the relevance of microglial origin on morphological and functional aspects of microglial cultures. The results underline that the tissue source for microglial cultures impacts the obtained results in different methods and is therefore an important note in the field of microglia research.
The authors thoroughly described the experimental methods and results and discussed the obtained results in a detailed and reasonable fashion. However, some pieces of important information are missing to evaluate the research design:
Major remarks:
The authors refer to supplemental material which is, unfortunately, missing in the manuscript version that I received. I kindly ask the authors to add the supplementary material.
Specifically, I am uncertain which cell populations were quantified in figure 1 D-K. The axis labels state ‘of total cells’. Is this the total number of events, single cells, live cells or CD11b+ cells? Please specify.
Please add how many biological replicates were used for the study. How many animals were used for microglial cultures and for histological analysis?
Minor remarks:
Please provide larger magnifications of fluorescent images in figure 1B, 3B and 3E for better evaluation of the staining quality and microglial morphology.
Please check the axis labels in figure 1D.
Please check line 338 which states that white matter microglia have been used for histological analysis.
Round 2
Reviewer 1 Report
Accept with modification
Reviewer 2 Report
The author's have thoroughly adapted the manuscript in regard to all raised concerns. I would like to express my appreciation for the accurate point-to-point response and the high-quality graphical presentation style of the results. I recommend publication of the manuscript.